


# Land Subsidence due to groundwater pumping: Hazard Probability Assessment through the Combination of Bayesian Model and Fuzzy Set Theory

Huijun Li[1], Lin Zhu[1], Gaoxuan Guo[2], Yan Zhang[3], Zhenxue Dai[4], Xiaojuan Li[1], Linzhen Chang[5], Pietro Teatini[6,7]

[1] Laboratory Cultivation Base of Environment Process and Digital Simulation, Beijing Laboratory of Water Resources Security, Key Laboratory of 3-Dimensional Information Acquisition and Application, Capital Normal University, Beijing, 100048, China

[2] Beijing Institute of Hydrogeology and Engineering Geology, Beijing, China

[3] Key Laboratory of Earth Fissures Geological Disaster, Ministry of Natural resource,Geological Survey of Jiangsu Province, Jiangsu, China

[4] College of Construction Engineering, Jilin University, Changchun 130026, China;

[5] Fourth Institute of Hydrogeology and Engineering Geology, Hebei Geology and Mineral Exploration and Development, Hebei, China

[6] Dept. of Civil, Environmental and Architectural Engineering, University of Padova, Padova 35121, Italy

[7] UNESCO-LaSII (Land Subsidence International Initiative), Querétaro, Mexico

*Corresponding to: Lin Zhu, hi-zhulin@163.com*

**Abstract** Land subsidence caused by groundwater over-pumping threatens the sustainable development in Beijing. Hazard assessments of land subsidence can provide early warning information to improve prevention measures. However, uncertainty and fuzziness are the major issues during hazard assessments of land subsidence. We propose a method that integrates fuzzy set theory and weighted Bayesian model (FWBM) to evaluate the hazard probability of land subsidence measured by Interferometric Synthetic Aperture Radar (InSAR) technology. The model is structured as a directed acyclic graph. The hazard probability distribution of each factor triggering land subsidence is determined using Bayes' theorem. Fuzzification of the factor significance reduces the ambiguity of the relationship between the factors and subsidence. The probability of land subsidence hazard under multiple factors is then calculated with the FWBM. The subsidence time-series obtained by InSAR is used to infer the updated posterior probability. The upper and middle parts of the Chaobai River alluvial fan is taken as a case-study site, which locates the first large-scale Emergency Groundwater Resource Region in Beijing plain. The results show that rates of groundwater level decrease larger than 1 m/y in the confined and unconfined aquifers, compressible layer thicknesses between 160 and 170 m, and Quaternary thicknesses between 400 and 500 m yield maximum hazard probabilities of 0.65, 0.68, 0.32, and 0.35, respectively. The overall hazard probability of land subsidence in the study area decreased from 51.3% to 28.3% between 2003 and 2017 due to lower rates of groundwater level decrease. This study provides useful insights for decision-makers to select different approaches for land subsidence prevention.



## 1. Introduction

The continuous over-pumping of groundwater results in dramatic piezometric drawdown and induces regional land subsidence. Many countries such as China, Mexico, Italy, USA, Spain, Iran (Teatini et al., 2005; Tomás et al., 2010; Galloway and Burbey, 2011; Chaussard et al., 2014; Zhu et al., 2015; Motagh et al., 2017), has reported the land subsidence due to groundwater pumping. Land subsidence is a complex process influenced by the anthropogenic activities and geological environment. The anthropogenic extraction of groundwater from aquifer is the principal triggering factor because the rapid decline in the

groundwater level leads to the compaction of the aquitard, and consequently, the land surface subsides (Xue et al., 2005; Zhu et al., 2015; Gao et al., 2018). Although the drops of groundwater level in aquifers lead to the land subsidence, but this process is also controlled by the geological environment which includes hydrologic and geomechanic conditions (Zhu et al., 2015, 2017; Gambolati and Teatini, 2015). Terzaghi's effective stress principle shows that a decrease in the pore pressure leads to an increase in the effective stress which consequently induces land subsidence, and this process is related to the soil mechanical

properties (Bonì et al., 2020). Land subsidence threatens the environment and cause economic losses, such as municipal infrastructure damage, building fracture and increasing flood risk (Wu et al., 2017; Peduto et al., 2017; Wang et al., 2018). Assessments of the subsidence hazard are necessary for risk prevention.

Recent studies have analyzed the hazards of land subsidence to buildings using field investigation and Interferometric Synthetic Aperture Radar (InSAR) technology (Julio-Miranda et al., 2012; Tomás et al., 2012; Bhattarai et al., 2017; Peduto et al., 2017).

Some studies assessed the regional subsidence hazard and identified the high-risk area using spatial modelling method with GIS (Huang et al., 2012; Bhattarai et al., 2017), multi objective decision making (Jiang et al., 2012; Yang et al., 2013), and advanced methods along with fuzzy set theory (Tafreshi et al., 2019). These methods are usually subjective and qualitative. Land subsidence is a geological problem with various random natural variables. Hazard assessment is associated with an inherent degree of uncertainty, which includes aleatoric aspects due to randomness and epistemic aspects related to insufficient

information (Kiureghiana and Ditlevsen, 2009). Aleatoric uncertainty may come from the randomness of natural variables, the validity of the data (Matthies, 2007). Epistemic uncertainty may be generated by inadequate expert knowledge and the selection of evaluation factors and their quantitative effects on a hazard (Vilares and Kording, 2011). The methods mentioned above do not fully consider these uncertainties.

To avoid these disadvantages, some researchers have adopted more objective methods, such as evidence reasoning methods

(Chen et al., 2014; Pradhan et al. 2014), numerical models based on the physical mechanism (Xu et al., 2015; Dai et al., 2016; Jia et al., 2018; Sundell et al., 2019), and machine learning (Park et al., 2012; Yi et al., 2017). However, numerical models require detailed geo-hydrological and geological parameters, which are difficult to collect for model initialization (Smith and Knight, 2019). Evidence reasoning has strict combination rules and becomes exponentially intensive from the computational point of view as the number of elements increases, although it can handle both certain and uncertain information regardless of

whether the information is complete or incomplete and precise or imprecise (Dai, 1999). Furthermore, the current studies are mainly focus on the identification and classification of hazard level without any quantitative analysis of the subsidence hazard or the hazard of single factor.

The main challenges in the field are to reduce the uncertainty of hazard assessments and to find an objective and effective method to assess hazard areas and risks. The mentioned uncertainty can be represented with probabilities. Bayesian models

(BMs) are powerful probability approaches to deal with uncertainty (Vilares and Kording, 2011). BMs have been widely applied in disaster hazard assessment, such as flood hazard and pipeline damage assessments (Liu et al., 2017; Zhang et al. 2016).

This paper proposes a fuzzy weighted BM (FWBM) that combines a weighted BM (WBM) and fuzzy set theory to evaluate the subsidence hazard probability and analyze the hazard probability for different rates of groundwater level change. The

posterior probability was calculated using InSAR-derived land subsidence as the model input to reduce the epistemic uncertainty. This new approach is applied in the Chaobai River alluvial fan in Beijing, China, which locates the first large-scale Emergency Groundwater Resource Region (EGRR) that supplies water to Beijing. The hazard probability inferenced with the proposed relatively objective method can offer scientific support for land subsidence early warning and prevention.

## 2. Methodology

### 2.1 InSAR technology


InSAR is a microwave remote sensing technique that records the phase and amplitude of the electromagnetic waves of ground objects. The phase information is used to inversely determine the subsidence. Persistent Scattered InSAR (PS-InSAR) is the most popular technology for detecting time series of subsidence by calculating the differential interferometric phase of PS points with a detection accuracy of millimeters (Sun et al., 2017). The density of PS points can reach $450/km^2$ in urban areas

(Ferretti et al. 2011). The differential interferometric phase Φ of each PS in the corresponding interferogram contains five components: the deformation phase along the line of sight (LOS), the topographic phase, the phase component due to the atmospheric delay, the orbital error phase, and the phase noise (Teatini et al. 2007). The deformation phase along the LOS can be extracted by removing other phase information.

PS-InSAR technology includes four steps:

(1) master image selection

(2) construction of a series of interferograms

(3) PS point selection

(4) unwrapping phase

### 2.2 FWBM method



### 2.2.1 Basic principle of BM

BMs consider the probability distributions of random variables and can infer the posterior probability based on weakly informative prior probability to address uncertainty (Weise and Woger 1993). A BM consists of a set of random variables with complex causalities that can be plotted using a directed acyclic graph (DAG), where random variables are represented as eigenvector nodes (Ren et al. 2009). In DAG (Fig. 1(a)), the hazard factors related to land subsidence are parent nodes ($Y_j$), and the subsidence hazard is the child node ($T$). The arrows represent the probabilistic dependence between nodes (Korb and Nicholson 2003).

Bayes' theorem can be used to infer posterior probability distributions from weakly informative prior probability distributions through observed results (Verdin et al. 2019). The approach is formulated as follows:

$$P(Y|S) = \frac{P(Y)P(S|Y)}{P(S)} \tag{1}$$

where $S$ represents the observed land subsidence; $P(Y|S)$ is the posterior probability of $Y$ subject to $S$; $P(Y)$ is the prior probability independent of $S$; $P(S|Y)$ is the likelihood function, representing the development of $Y$; and $P(S)$ is the marginal probability.

For multiple factors in DAG, the jointly probability of multiple conditions can be expressed as

$$P\{T|Y_1, \ldots, Y_j, \ldots, Y_m\} = \prod_{j=1}^{m} P(T|Y_j) \tag{2}$$

where $Y_j$ is the $j$-th factor that influences $T$.

### 2.2.2 FWBM construction

The conditional independence assumption must be met for BMs. This assumption generally can be strictly met in geological studies (Webb and Pazzan 1998). WBMs use weighted assessment variables to relax the independence assumption and address the different contributions of parent nodes to child nodes (Webb and Pazzan 1998). It has been widely used in hazard-related analyses (Tang et al. 2018). However, the weight of each factor is determined by its importance to land subsidence, which is usually qualitative and fuzzy, such as decline of piezometric head being the main driver of subsidence, compaction of a compressible layer, or high static loads influencing subsidence (Chen et al., 2016; Li et al., 2017). The fuzziness of the contribution of the factors to land subsidence may cause ambiguity in weighting when determining the importance of the factors. These deviations can be modeled with fuzzy set theory which can express fuzziness through a membership function to objectively describe the relationship between land subsidence and the factors (Mentes and Helvacioglu 2011). Therefore, the fuzzification of factor importance is applied to eliminate ambiguity. With the fuzzy-based weight, the FWBM is constructed with the following equation, which is extended from Equation (2), to determine the probability of random variable $T$:

$$P\{T|Y_1, \ldots, Y_j, \ldots, Y_m\} = \prod_{j=1}^{m} P(T|Y_j)^{w_{Fj}} \tag{3}$$





where $w_{\mathrm{F}j}$ is the fuzzy-based weight of $Y_j$.

The structure of the FWBM is shown in Fig. 1(b), which is an improvement of Fig. 1(a). The eigenvector nodes are fuzzy weighted.

As spatial variables, the hazard probability of factors is calculated through its spatial features using BM. The spatial features of $Y_j$ are given by $X$, $X=\{X_{j,1}, X_{j,2}, \dots, X_{j,i-1}, X_{j,i}\}$, where $X_{j,i}$ is defined as the $i$-th feature of the $j$-th factor, as shown in Fig. 1(c). The value of $i$ depends on the feature classification. Obviously, FWBM contains three parts, probability of $Y_j$ which is

consists of its spatial feature $X_{j,i}$, probability of $T$,

The hazard probability of spatial feature $X_{j,i}$ at subsidence detection time $k$ is calculated using the following equation, which is derived from Equation (1):

$$P\{X_{j,i}|S,\ t=k\} = P(X_{j,i}, t=k)\frac{P\{S|X_{j,i}, t=k\}}{P(S)} \tag{4}$$

where $P(X_{j,i}, t=k)$ is the prior probability, with the initial value calculated with the feature grid number ratio;

$P\{S|X_{j,i}, t=k\}$ is the conditional probability calculated with the ratio of the subsidence grid to the feature grid; and $P(S)$ is the marginal probability, which is the sum of the probability of $X_{j,i}$ and is calculated by the following equation:

$$P(S) = \sum_{i=1}^{n} P(X_{j,i}, t=k)P\{S|X_{j,i}, t=k\} \tag{5}$$

### 2.2.3 FWBM implementation

In the FWBM framework (Fig. 2), a BM is used to infer the hazard probability with the fuzzification of factor importance to

reduce the ambiguity of the relationship between hazard factors and land subsidence based on subsidence data obtained with InSAR technology.

The first part is data processing to obtain the standardization dataset. The assessment hazard factors are derived first which are parent nodes $(Y_1, Y_2 \dots Y_m)$ in the model structure from groundwater extraction and geological conditions. Additionally, the posterior probability in a BM can be adjusted by using the new observed events to reduce the epistemic uncertainty (Weise

and Woger 1993). For assessments of land subsidence hazard, InSAR technology can be applied to obtain time series of land subsidence at regional scale. Therefore, the stack of SAR images are processed with PS-InSAR to obtain the time series of PS points that provided the subsidence information. The PS points form a continuous input dataset to update the posterior probability of the FWBM, thus reducing the uncertainty in the assessment process. Simultaneously, the subsidence data is also used to validate the hazard assessment outcome. Then, these two datasets are standardized into grid form and spatially

connected using the spatial join tool in GIS for the statistical analysis of FWBM.

The second part is the model implementation, which contains three modules corresponding to the three variables that should be inferred in FWBM, that is probability of $Y_j$ and $T$, factor weight $w_{\mathrm{F}j}$.

- The first module is inferring the subsidence hazard probability of $Y_j$, $P(Y_j)$. For a single factor, the posterior probability


distribution of subsidence hazard is inferred through its spatial feature $X_{j,i}$ using the Bayesian theorem. As shown in Fig. 3(a),

the hazard probability of $X_{j,i}$ $P\{X_{j,i}|S, \ t=k\}$ is calculated using Equation (4) and Equation (5) with the calculated prior probability $P(X_{j,i}, t=k)$ and conditional probability $P\{S|X_{j,i}, t=k\}$ at subsidence detection time $k$. The prior probability and conditional probability are calculated through spatial statistical analysis with the land subsidence data. This step is iterated when new subsidence event was observed (new PS points detected and as input) to update the posterior probability.

- The second module is calculating the fuzzy-based weight $W_{Fj}$. Fuzzification of the factor importance is processed by

establishing fuzzy pairwise comparison matrices (f_PCM). According to the analytic hierarchy process (AHP) method, the pairwise comparison criteria was divided into five levels represented with odd numbers from 1-9 (Saaty 1980). The five levels were regarded as fuzzy numbers, and the medium level was considered to be equally important. The value of each level was expressed as a triangular fuzzy number which is commonly used to express fuzziness (Mentes and Helvacioglu 2011). Based on the constructed f_PCM, $W_{Fj}$ was calculated by the fuzzy extended AHP method (Van and Pedrycs 1983).

- The third module is inferring the probability of $T$, $P(T)$. The hazard probability influenced by multiple factors is derived using the FWBM. As shown in Fig. 3(b), with the probability density $P(Y_j)$ and factor weights, the gridded hazard probability of land subsidence $P(T)$ is implemented using Equation (3). The hazard probability map is reclassified using the natural breaks (Jenks) classification method, which is widely used in risk evaluation (Suh et al. 2016; Liu et al., 2017), and compared with the InSAR detected subsidence to validate the assessment results.

**3.   Case study**

**3.1   Description of the study area**

The study area belongs to the upper-middle part of the Chaobai River alluvial fan in the northern Beijing plain and covers approximately 1,350 km² (Fig. 4). The Huairou EGRR is located in this area and was designed to ensure the urban water supply in continuous dry or emergency conditions. Long-term groundwater over-pumping has caused rapid decreases in the

groundwater level, with a maximum value of approximately 40 m after the operation of the EGRR in 2003 (Zhu et al., 2015, 2016). This significant drop resulted in regional land subsidence. To relieve the situation, the South-to-North Water Transfer Project-Central Route (SNWP-CR) was implemented at the end of 2014.

**3.2   Datasets and processing**

In this study, the rates of groundwater level change in confined ($Y_1$ with the feature expressed as $X_{1,i}$, $i=1\ldots4$) and unconfined

($Y_2$ with the feature expressed as $X_{2,k}$, $k=1\ldots4$) aquifers which reflecting the groundwater drawdown, the cumulative thicknesses of the compressible layers ($Y_3$ with the feature expressed as $X_{3,m}$, $m=1\ldots17$), and the thickness of the Quaternary unit ($Y_4$ with the feature expressed as $X_{4,n}$, $n=1\ldots14$) which reflecting the geological conditions were chosen as hazard factors. The four factors are classified according to the data characteristics and mapped in Fig. 5.



The datasets of factors cover three periods:

(1) From January 2003 to December 2010, a period of massive groundwater exploitation after the operation of the EGRR.

(2) From January 2011 to December 2014, the rate of decline in the groundwater level slowed due to the long-term loss of groundwater and increased rainfall (Zhang et al. 2015).

(3) From January 2015 to December 2017, operation of the SNWP-CR partially relieved groundwater exploitation.

A total of 125 SAR images were collected, including 37 ASAR images from June 2003 to January 2010, 38 RADARSAT-2

images from November 2010 to November 2014, and 50 Sentinel-1 images from December 2014 to December 2017. The subsidence results were validated and calibrated with an extensometer station and benchmark data (the location is shown in Fig. 4), with an error of ±7 mm (Zhu et al. 2015, 2020a). The PS points with a subsidence rate above 10 mm/y were regarded as subsidence points.

The study area was gridded into 5,664 cells, with a cell size of 500 m × 500 m. Grid sizes of 200 m, 500 m, and 1000 m were

compared. The assessment results for the 200 m grid size were similar to the results for the 500 m grid size with less smooth edges but a higher computational cost. The results for the 1000 m grid size displayed a low resolution. All factors and subsidence data are connected to the grid and each grid ID has five features including four assessment factors and subsidence.

### 3.3 Model implementation

### 3.3.1 Weight computation

The fuzzification of factor importance is expressed as a triangular fuzzy number considering the ambiguity between factors and subsidence. The fuzzy-based weights ($W_{Fj}$) of the factors are shown in Table 1. To compare the model performance when ambiguity was eliminated, we implemented the WBM with the non-fuzzy-based weight ($W_j$) calculated by the AHP method.

**Table 1 Fuzzy ($W_{Fj}$) and non-fuzzy-based ($W_j$) weights for the hazard factors**

|          | $Y_1$ | $Y_2$ | $Y_3$ | $Y_4$ |
|----------|-------|-------|-------|-------|
| $W_{Fj}$ | 0.32  | 0.12  | 0.38  | 0.18  |
| $W_j$    | 0.33  | 0.10  | 0.43  | 0.14  |

### 3.3.2 Probability of $Y_j$ inference

The hazard probability of feature $X_{j,i}$ P$\{X_{j,i}|S, t = k\}$ is calculated first. For example, $X_{3,12}$ is the twelfth feature of the compressible layer thickness ($Y_3$), indicating that the thickness is between 160 m and 170 m. The prior probability P$(X_{3,12})$ was calculated based on the ratio of the feature grid number to the total number of grids in the study area. The conditional probability P$\{S|X_{3,12}, from\ 2003\ to\ 2010\}$ is the percentage of grid cells for which subsidence occurred in feature $X_{3,12}$ from 2003 to 2010, which is used as input for the FWBM. P$\{S\}$ is the sum of P$\{S|X_{3,12}, from\ 2003\ to\ 2010\}$ calculated

based on Equation (5). The posterior hazard probability P$\{X_{3,12}|S, from\ 2003\ to\ 2010\}$ is calculated using Equation (4).

When the subsidence data from 2011 to 2014 or from 2015 to 2017 are used as input, this step is conducted in the same way



that it was for the period from 2003 to 2010, and the posterior hazard probability at a previous time is set as the prior probability at the current time.

3.3.3 Probability of *T* inference

With the probability of single factor and factor weights, the hazard probability of *T*, P(*T*) is then calculated. Since there was no confined aquifer and the subsidence was relatively low in the northern part of the study area, the hazard probability in these areas were set as 0.01.

## 4. Results and discussion

### 4.1 Validation of the results

The proposed FWBM was successfully applied to assess the subsidence hazard probability in the upper and middle part of the Chaobai River alluvial fans from 2003 to 2017. The hazard assessment distribution was reclassified into 7 grades (Fig. 6(a)). A hazard probability less than 0.07 indicates a low hazard region, and a hazard probability greater than 0.15 indicates a high hazard area (Fig. 6(b)).

The changes in the land subsidence rate (Sr) detected by InSAR (Fig. 6(c)) between 2010-2014 and 2015-2017 were utilized

to validate the assessment results. A positive value means the subsidence rate decreased (SrD), and a negative value means the subsidence rate increased (SrI). The total match ratio is 85% (Table 2). Notably, the reason that SrD was located in the high hazard region is that the piezometric level continuously decreased and the thick of compressible layers is large, additionally this area had high subsidence rate larger than 50mm/y as of 2017 (Zhu et al. 2020a, b).

### 4.2 Comparison of the FWBM and the WBM

The FWBM assessment results were compared with the results from a WBM that ignored the ambiguity in the hazard assessment framework. The WBM results were also divided into 7 levels. The levels of change between them were calculated (Fig. 7(a)); a negative value means the WBM had a higher hazard level, and a positive value means the WBM had a lower hazard level than the FWBM. In terms of the subsidence rate change (Fig. 6(c)), the WBM overestimated the subsidence hazard level for area 1 (Fig. 7(b)) and partially overestimated the level for area 2 (Fig. 7(c)) for subsidence rate decreases in these

areas. In addition, the WBM underestimated the hazard level for area 3 (Fig. 7(d)), where the subsidence rate increased. The FWBM performed better in regions with SrD and had a higher total match ratio than the WBM, as shown in Table 2.

**Table 2. Comparison of the match ratio obtained with FWBM and WBM**

|  | FWBM | | WBM | |
| --- | --- | --- | --- | --- |
|  | Number of SrI | Number of SrD | Number of SrI | Number of SrD |
| High hazard area | 1473 | 189 | 1497 | 274 |
| Low hazard area | 199 | 766 | 175 | 681 |
| Percentage | 88% | 80% | 89% | 71% |





| Total match ratio | 85% | 82% |
|---|---|---|

**4.3  Effect of assessment factors on hazard probability**

The hazard probability of assessment factors is shown in Fig. 8. Taking the period from 2015 to 2017 as an example, for the

rate of groundwater level change in the confined aquifer, a rate reduction greater than 1 m/y has a maximum hazard probability

of 0.65. The situation is the same for the rate of groundwater change in the unconfined aquifer; the maximum hazard probability

is 0.68 when the rate reduction exceeds 1 m/y. The results show that the higher reduction rate of groundwater level, the higher

hazard probability, this is consistent with previous studies which revealed that the rapid decline in the groundwater level leads

to the land surface subside (Tomás et al., 2010; Galloway and Burbey, 2011; Zhu et al., 2015). Compressible layer thicknesses

between 160 m and 170 m yield a maximum hazard probability of 0.32, and Quaternary thicknesses between 400 m and 500

m yields a maximum hazard probability of 0.35. This is consistent with other studies which showed that subsidence mainly

occurred over the area where the compressible layer thickness exceeds 100 m (Lei et al. 2016).

**4.4  Temporal change in subsidence hazard**

Because land subsidence is negligible in the north part of the study area (Zhu et al. 2015), where there is a uniform unconfined

sandy gravel layer with a small occurrence of compressible soils, the regions with confined aquifers were used to analyze the

temporal change in the hazard probability of land subsidence. As shown in Fig. 6(a), the subsidence hazard probability in

Niulanshan decreased from 2003 to 2017. However, the southwest part always maintains a high hazard probability, especially

in Tianzhu and Nanfaxin, where groundwater level changes exceeded 1 m/y and the thickness of the cumulative compressible

sediments exceeds 150 m.

Fig. 9 shows that the subsidence hazard probability value was between 0.01% and 51.30% from 2003 to 2010, between 0.01%

and 45.54% from 2011 to 2014, and between 0.01% and 28.33% from 2015 to 2017.

Overall, the subsidence hazard decreased. This should be credited to the implementation of water resource exploitation and

utilization policies. From 2003 to 2010, the operation of the EGRR led to rapid drawdown. From 2015 to 2017, the operation

of the SNWP-CR conveyed a large amount of water to Beijing, reducing the pressure on the groundwater and slowing the rate

of change in the groundwater level. Notably, the California State Water Project which also is a large water-transfer system has

been fundamental to control land subsidence (Zhu et al., 2020b).

**4.5  Spatial distribution of subsidence hazard**

Four subsidence hazard levels were classified from the probability map (Fig. 6(b)), consistent with previous studies (Yang et

al. 2013; Zhu et al. 2015). The high hazard covered 10.7% of the total area, and the medium hazard accounted for 17.5% of

the total area. The low and very low hazards represented 29.7% and 42.1% of the total area, respectively. As the thickness of

the compressible sediments increases from the north to the south, the subsidence hazard probability increased accordingly.

Tianzhu, Nanfaxin, Gaoliying, and Houshayu in the southwestern region experienced medium-high hazards because the compressible strata in these areas are thick and the groundwater table dropped significantly. The InSAR results also revealed that the maximum subsidence rate in these regions increased to 84.9 mm/y from 2015-2017. Overall, the area of high
subsidence hazard decreased due to the reduction in the rate of groundwater level change.

## 5.    Conclusions

Considering the ambiguity of the importance of various factors controlling land subsidence due to groundwater pumping and the uncertainty in the assessment process, the FWBM model was constructed to assess the probability of land subsidence hazards at a regional scale by combining a BM and fuzzy set theory. The InSAR technology was used to obtain land subsidence
time series to adjust the posterior probability of the FWBM thus reducing the model uncertainty.

The implementation of the FWBM in the Beijng area demonstrated the potentiality of this modelling approach and showed that it is superior when the ambiguity of the relationship between the factors and the subsidence is considered. The study is a first analysis of the hazard probability of land subsidence and the related hazard factors. From the case study, we found that subsidence probability decreased over time at three periods due to change of water utilization, such as the operation of the
SNWP-CR. When groundwater level reduction rates are greater than 1 m/y in the unconfined and confined aquifers, it yields maximum hazard probabilities of 0.68 and 0.65, respectively. When compressible layer thicknesses are between 160 m and 170 m, it yields a maximum hazard probability of 0.32. When Quaternary strata thicknesses are between 400 m and 500 m, it yields a maximum hazard probability of 0.35. The overall subsidence hazard probability of the study area decreased from 51.3% to 28.3% between 2003 and 2017 due to the decrease in the groundwater level reduction rate.

The results of this study suggest that the proposed subsidence hazard assessment method significantly represents the uncertainty and ambiguity compared to traditional qualitative methods (Huang et al., 2012; Park et al., 2012; Yang et al., 2013; Chen et al., 2014; Tafreshi et al., 2019; Sundell et al., 2019). The hazard probability map of different time period with different groundwater level conditions can offer scientific support for land subsidence early warning and help stakeholders and decision-makers to develop more reliable water utilization strategies taking into account the land subsidence hazards.

The prior probability in this model is determined by the factor grid number ratio, which may have deviations. This ratio can be further improved by expert knowledge. Additionally, the impact of the selected assessment factors on the results may be discussed in our next-step study. The subsidence conditions that reflecting the severity of subsidence (such as cumulative subsidence and subsidence rate) should be considered to assess the subsidence hazard in the future from a more comprehensive perspective.

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

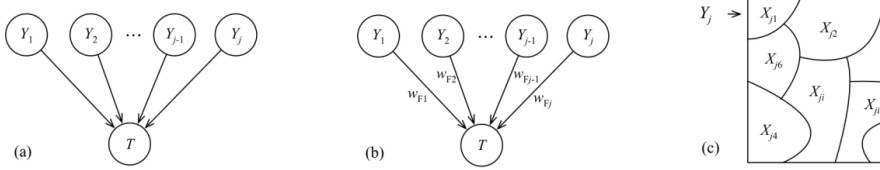

**Fig. 1 (a) DAG of the BM structure; (b) FWBM structure; (c) spatial features of the hazard factors ($Y_j$, for example)**




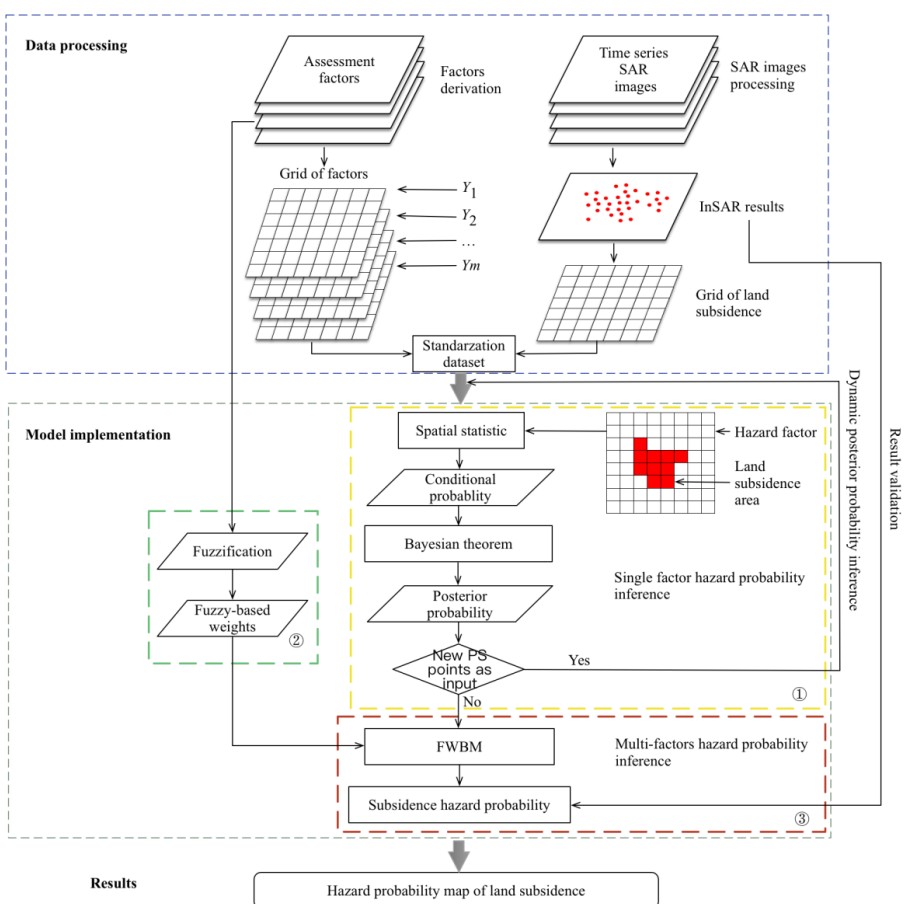

Fig. 2. Flowchart of subsidence hazard assessment using the FWBM

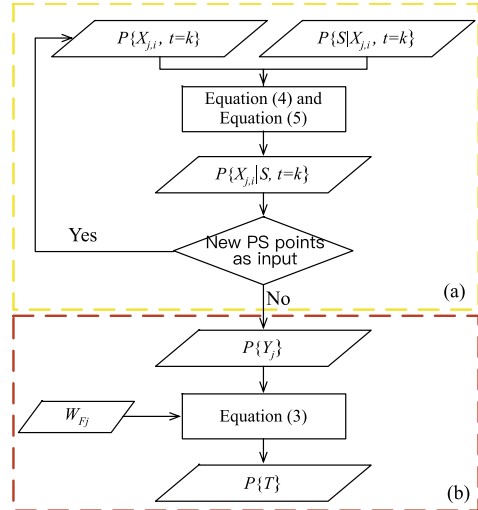

Fig. 3. Flowchart of infer (a) subsidence hazard probability of a single factor and (b) subsidence hazard probability influenced by multiple factors



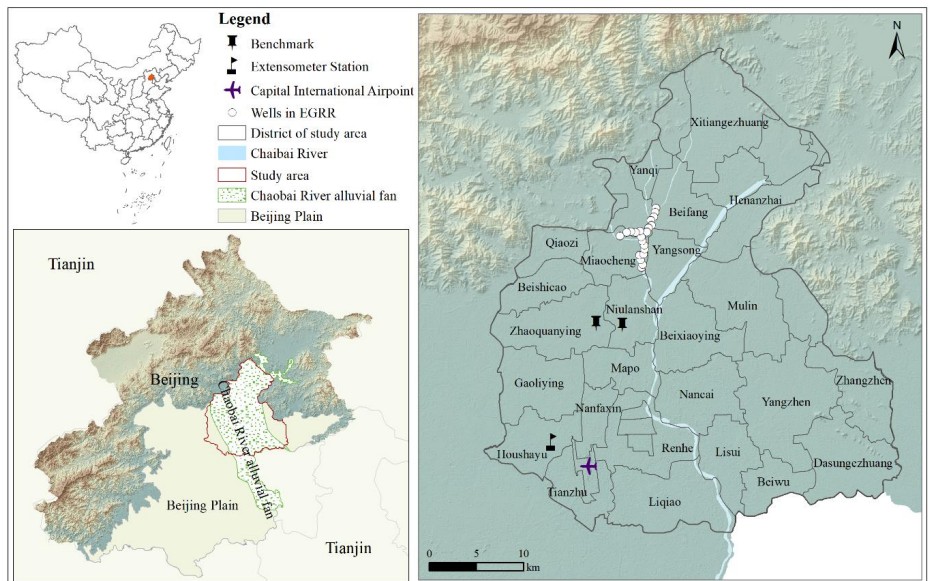

**Fig. 4. Location of the study area (the digital elevation model data is from the Shuttle Radar Topography Mission - SRTM -database; the administrative map is from the Beijing Institute of Geo-Environment Monitoring)**




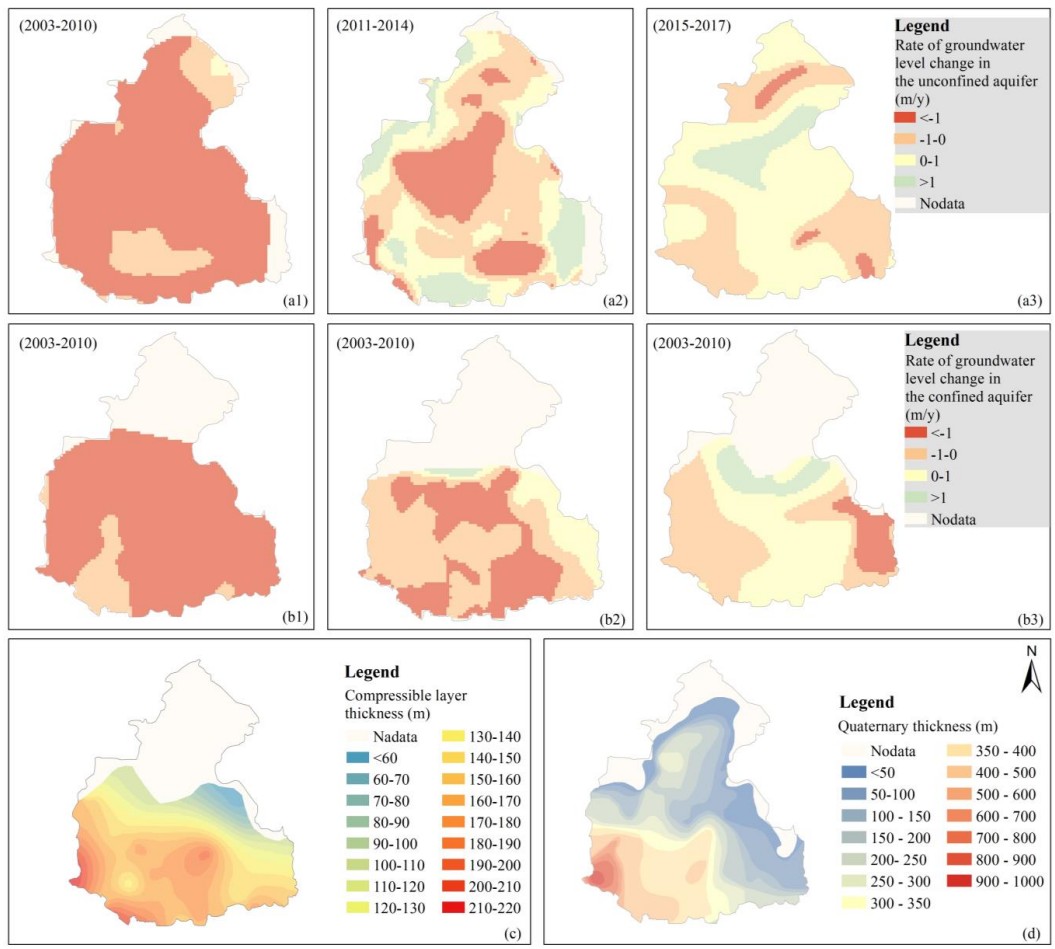

**Fig. 5. Assessment factors: (a1-3) Rate of groundwater level change in the unconfined aquifer (2003-2010, 2011-2014, 2015-2017, respectively. Negative values mean lowering); (b1-3) Rate of groundwater level change in the confined aquifer system (2003-2010, 2011-2014, 2015-2017, respectively. Negative values mean lowering); (c) Compressible layer thickness; (d) Quaternary thickness**

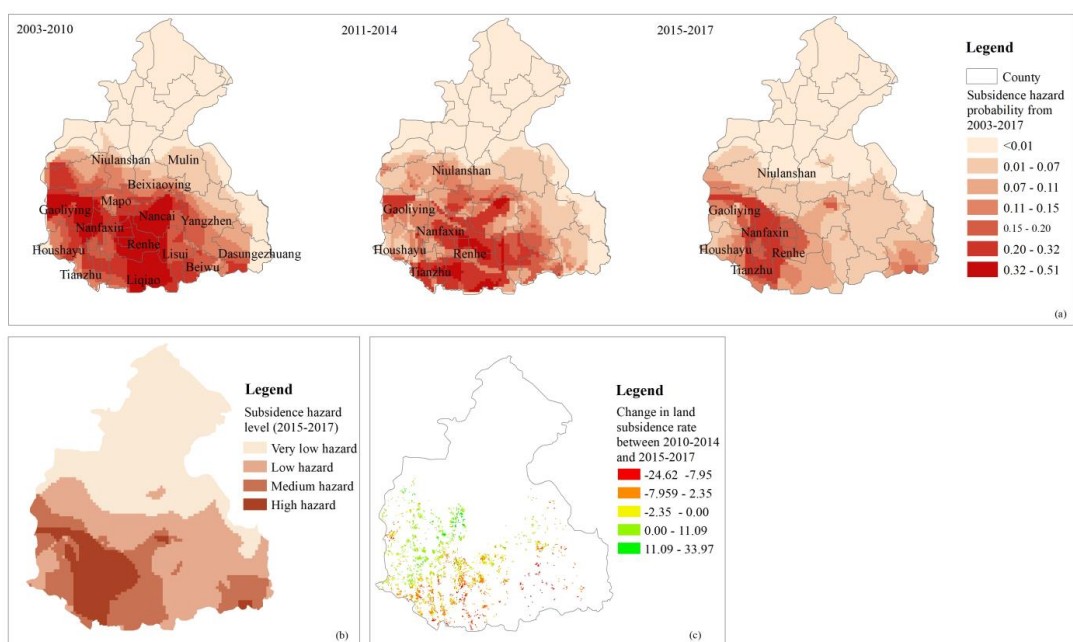

**Fig. 6. (a) Assessment of the subsidence hazard probabilities from 2003 to 2010, 2011 to 2014, and 2015 to 2017; (b) Subsidence hazard level (2015-2017); (c) Change in land subsidence rate between 2010-2014 and 2015-2017 obtained by InSAR**

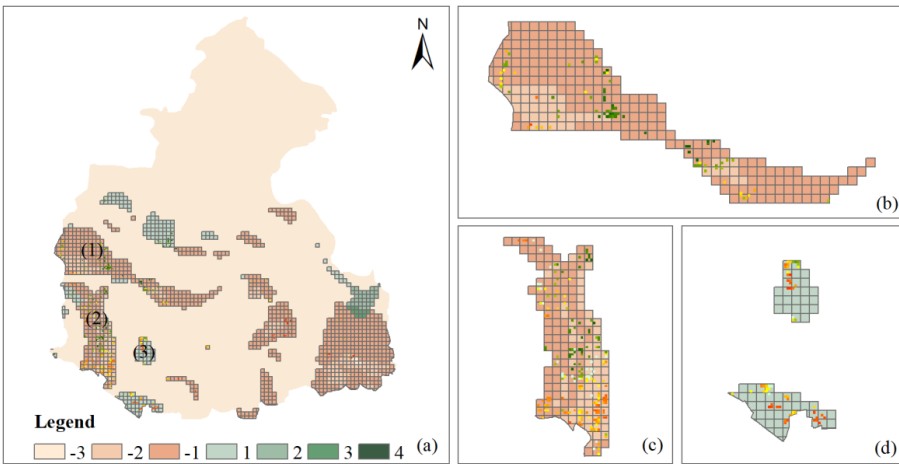

**Fig. 7. (a) The subsidence hazard level of change between the FWBM and the WBM (a negative value means the WBM had a higher hazard level, and a positive value means the WBM had a lower hazard level than the FWBM). An amplification of areas (1), (2), and (3) highlighted in (a) is provided in (b), (c), and (d), respectively. The colored dots represent the subsidence rate same as figure 6(c) (the green dots represent the decreased subsidence rate which means a lower hazard probability, the red dots mean a higher hazard probability)**





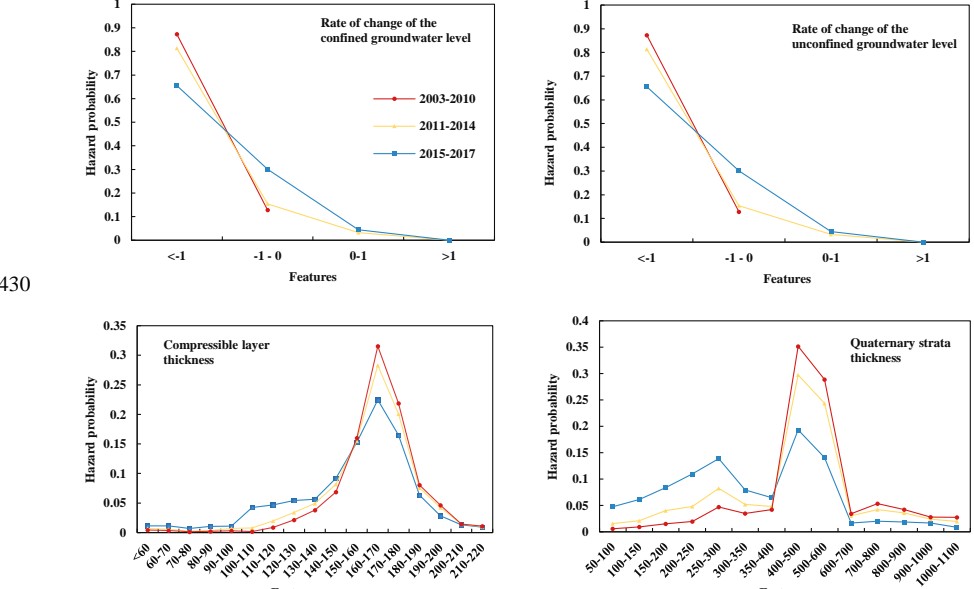

**Fig. 8. Hazard probability of factors for the three time periods from 2003 to 2017 of the study area**

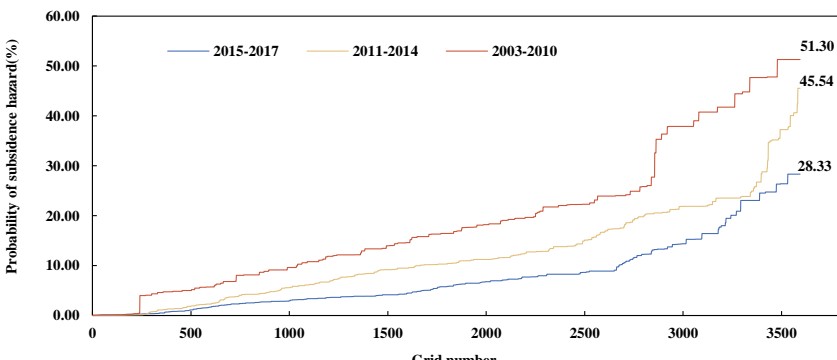

**Fig. 9. Probability distribution of subsidence hazard for the three time periods from 2003 to 2017 of the study area**