# Peer review of "Land Subsidence due to groundwater pumping: Hazard Probability Assessment through the Combination of Bayesian Model and Fuzzy Set Theory"

_Natural Hazards and Earth System Sciences, 2020_

## Referee Comment (RC1) · Anonymous Referee #1 · 18 Dec 2020

Dear Editor, I would like to thank you for inviting me as a reviewer for the paper "Land Subsidence due to groundwater pumping: Hazard Probability Assessment through the Combination of Bayesian Model and Fuzzy Set Theory" by Huijun Li et al. I really appreciated the authors' approach on land subsidence topic, and I hope my comments would open an interesting and constructive discussion with the authors to improve the paper.

General comments

In this paper the authors proposed an interesting method to evaluate hazard probability of land subsidence in the Beijing Plan, by integrating Bayesian Model and Fuzzy

Set Theory. By using InSAR data, groundwater levels and geological characteristics of the area, authors analysed, modelled and validated the changes in land subsidence rates from 2003 to 2017. They also highlighted the differences in the resulting maps, comparing models with and without fuzzification of the factors' weights, which provided interesting findings related to the importance of the characterization of uncertainties when dealing with the subsidence phenomenon. Addressing uncertainties in approaching natural hazards is of primary importance and necessity, as well their subsequent communication to the administrations in charge. The introduction section provides an extensive framework of the subsidence topic in relationship with other works and approaches. However, it needs some aspects to be clarified, especially when authors point out their approach is quantitative while some previous cited papers are qualitative. The method section, despite its intrinsic complexity, is well presented. The workflows presented in Fig. 1, Fig.2 and Fig.3 are certainly useful for the interpretation. However, some statements should be revised to improve clarity. The case study section appears as a hybrid between method and results; in my opinion this should be avoided. This is the part for which I think more work should to be done. Some operational choices need to be further clarified and explained. Finally, I appreciated the results section as it is concise and very well subdivided in sub-sections according to the different aspects of authors' findings. In particular, the comparison between models with and without fuzzification is meaningful, which translates into the comparison between accounting -or not accounting- for the uncertainties of the influencing factors. Also, the comparison between the two modelling approaches with InSAR derived subsidence rate changes, in terms of overestimation and underestimation is a really appreciable result. However, I would suggest to change the heading of the section in Results and Discussion. In my opinion, the work is worthy of publication, unless the aspects listed in the specific comments are addressed.

Specific comments

Introduction Section

Line 52: You are pointing out that the method adopted in the papers from line 50 to 52 are subjective and qualitative. You should clarify why, especially for the paper of Tafreshi et al. (2019), in which fuzzy theory is used too. Alternatively, this part could also be added in the results section, as a possible discussion of your findings in relationship with previous literature.

Line 78: I do not understand why you stated that your procedure could be implemented in an early warning system. This statement is also repeated in the conclusion section without any further explanation. Do you mean implementing in the procedure future groundwater scenarios or something else? I think you should clarify this point, otherwise I would change it in "prediction".

Methodology Section

Line 89: It is not clear if the mentioned steps were addressed by the authors or if they were already available.

Line 123: Is equation (3) a new formulation by the authors or could it be associated to some references?

Line 149: Standardization has a precise meaning. I would recommend to clarify, otherwise change this word if the meaning is that you have just created homogeneous size grids for InSAR data and not further processed them.

Case Study Section

Line 181-182: Could you further add some characterization to the geological units mentioned? E.g. Soil type, origin etc..

Line 183: This part is crucial, as you are basing your work on calculating different weights for the different features of each factor. Thus, the criteria used for the classification of each factor should be further clarified and justified (i.e. each class represents a factor's feature with its own computed weight, consequently crucial for hazard probability calculation). In my opinion is not enough stating that "the four factors are classified

according to the data characteristics".

Line 192-193: You have set the threshold between subsiding and stable points at 10 mm/year. Usually, this choice should be based on standard deviations analysis of the subsidence rates in a stable area, otherwise, if this threshold came from previous works in the area, it should be mentioned. Moreover, the threshold you have chosen should be used to define colours in your map in Fig. 6c. Indeed, in this figure, you have chosen the green colour also for subsidence rates from 0 to 10 mm/years.

Table 1: It could be interesting if you would provide in the discussion section some considerations about the factors which resulted in the highest difference between the computed fuzzy and no-fuzzy weights.

Line 212: Does it means that, differently for 2011-2014 and 2015-2017, for the period 2003-2010 the prior and posterior probability are computed on the same InSAR data? Could these make a difference in the three resulting hazard probability maps?

Results section

Line 226: Is that right that SrD (i.e. subsidence rate decrease) is located in the region with the highest groundwater level decrease between 2011-2014 and 2015-2017? I would say that where the subsidence rate decreased between these two periods, the groundwater level was restored accordingly, thus the area is SrD, and in a lower hazard class for 2015-2017. If I misinterpreted what you wanted to communicate, I suggest clarifying this part by using shorter statements.

Line 245-247 and linked Fig. 8: Could you provide an explanation (also with the help of previous works) for why after a peak of hazard probability is reached both for compressible layer and Quaternary layer at a certain thickness values and then, for higher thicknesses, the hazard probability is much more lower? In my opinion, an interesting discussion on this could be done.

Conclusions Section

Line 288: As I have previously mentioned in the comments for the Introduction section, I would suggest clarifying the statement regarding the early warning system.

Technical Comments Line 37 and Line 41: The use of "the" before land subsidence should be avoided.

Line 41: The use of "but" in the statement is unnecessary.

Line 55: I would recommend changing "validity of the data" into "data quality".

Line 89: I would suggest changing "technology" into "processing".

Line 104-105: the meaning of Y is not specified. Moreover, I think there is a mistake in the statement: "P(Y|S) is the posterior probability of Y subject to S". S is land subsidence, thus it is the phenomenon for which you want to calculate the probability influenced by the factor Y.

Line 129-130: You stated that the FWBM method contains three parts, but you just enumerated one, so the statement is incomplete. Figures: measurement units in figure 6c and 8 are missing.

About the English language, I think it needs some improvements (singular/plural and punctuation need to be checked, tenses need to be homogenised). If possible, I suggest a professional proofreading.

---

## Referee Comment (RC2) · Anonymous Referee #2 · 4 Jan 2021

The authors' approach to land subsidence risk assessments that can provide early warning information to improve prevention measures is relevant and original. There are several Bayesian approaches which all give a weighted average of the predictive distributions. However, often they are not applied correctly, which can lead to false conclusions. In this work, the integration of fuzzy set theory and the Bayesian Weighted Model (FWBM) assessing the hazard probability of land subsidence measured by Interferometric Synthetic Aperture Radar (InSAR) technology is a great approach. The different maps comparing models with and without fuzzification provide obvious information for improved predictions by a combination of models. The introduction is explicit and detailed on sag about other works and approaches. The document structure is

quite clear, and in my opinion, the work deserves to be published. However, the author needs to revise the English grammar, which is confused in some sections. In this work, some operations need to be more explaining for a better understanding of this approach.

---

## Author Comment (AC1) · 4 Jan 2021

**Responses to the comments of Referee 1**

We appreciate the positive and constructive comments provided by the Editor and the Referee 1. We have revised the manuscript carefully by incorporating those comments and suggestions. The following are our point-to-point responses to the referee comments.

**About the general comments**

**(1) The case study section appears as a hybrid between method and results; in my opinion this should be avoided. This is the part for which I think more work should to be done.**

**Response:** Suggestion followed. The part related to "results" has been moved to Discussion section. The method description kept in Section 3 only describes in detail how the methodology provided in Section 2 (Methodology) is applied to the study area.

**(2) I appreciated the results section as it is concise and very well subdivided in sub-sections according to the different aspects of authors' findings. In particular, the comparison between models with and without fuzzification is meaningful, which translates into the comparison between accounting -or not accounting- for the uncertainties of the influencing factors. Also, the comparison between the two modelling approaches with InSAR derived subsidence rate changes, in terms of overestimation and underestimation is a really appreciable result. I would suggest to change the heading of the section in Results and Discussion.**

**Response:** Thanks for this positive comments. We changed the heading of Section 4.2 "Comparison of the FWBM and the WBM" to "Effect of fuzziness on model results", with the considering that this section mainly discussed the fuzziness of factor importance to factor weights, then to model accuracy.

**Specific comments**

**(1) Line 52: You are pointing out that the method adopted in the papers from line 50 to 52 are subjective and qualitative. You should clarify why, especially for the paper of Tafreshi et al. (2019), in which fuzzy theory is used too. Alternatively, this part could also be added in the results section, as a possible discussion of your findings in relationship with previous literature.**

**Response:** This has been clarified in the Introduction section. Recent studies have analyzed the hazards of land subsidence to buildings using field investigation and Interferometric Synthetic Aperture Radar (InSAR) technology (Julio-Miranda et al., 2012; Tomás et al., 2012; Bhattarai et al., 2017; Peduto et al., 2017). Some studies assessed the regional subsidence hazard and identified the high-risk areas using the spatial modelling method with GIS (Huang et al., 2012; Bhattarai et al., 2017), multi objective decision making (Jiang et al., 2012; Yang et al., 2013), and advanced methods along with fuzzy set theory (Tafreshi et al., 2019). These methods require expert score which is subjective and the produced risk level map is also qualitative. Tafreshi (2019) et al. adopted fuzzy functions to standardize parameters with different dimensions, which did not address the fuzziness of parameter importance.

**(2) Line 78: I do not understand why you stated that your procedure could be implemented in an early warning system. This statement is also repeated in the conclusion section without any further explanation. Do you mean implementing in the procedure future groundwater**

scenarios or something else? I think you should clarify this point, other- wise I would change it in "prediction".

**Response:** Corrected. "Early warning" has been changed to "prediction" in Introduction and Conclusion sections. The calculated hazard probability can be as a prior probability for the prediction of land subsidence hazard probability with different groundwater scenarios.

**(3)   Line 89: It is not clear if the mentioned steps were addressed by the authors or if they were already available.**

**Response:** The steps for PS-InSAR technology were already available. A reference has been added.

**(4)   Line 123: Is equation (3) a new formulation by the authors or could it be associated to some references?**

**Response:** Equation (3) is a new formulation given by the authors, which is extended from Equation (2). This has been described in line 122.

**(5)   Line 149: Standardization has a precise meaning. I would recommend to clarify, otherwise change this word if the meaning is that you have just created homogeneous size grids for InSAR data and not further processed them.**

**Response:** Corrected. The sentence has been rephrased. We created homogeneous size grids for the datasets. The word 'standardization dataset' is changed to 'gridded dataset' in line 142 and Fig. 2.

**(6)   Line 181-182: Could you further add some characterization to the geological units mentioned? E.g., Soil type, origin etc.**

**Response:** Land subsidence mainly occurred in compressible layers with fine deposits in study area. This time we considered the cumulative thickness of compressible layers.

**(7)   Line 183: This part is crucial, as you are basing your work on calculating different weights for the different features of each factor. Thus, the criteria used for the classification of each factor should be further clarified and justified (i.e. each class represents a factor's feature with its own computed weight, consequently crucial for hazard probability calculation). In my opinion is not enough stating that "the four factors are classified according to the data characteristics".**

**Response:** The criteria for classification of each factor are given and discussed. The contour lines of thicknesses of compressible layer and quaternary strata were collected. We used the same classification without interpolation, which may introduce error. The change of groundwater level varied from -4m to 6m between 2015 and 2017. The rates of groundwater level change were classified into four classes.

**(8)   Line 192-193: You have set the threshold between subsiding and stable points at 10 mm/year. Usually, this choice should be based on standard deviations analysis of the subsidence rates in a stable area, otherwise, if this threshold came from previous works in the area, it should be mentioned. Moreover, the threshold you have chosen should be used to define colours in your map in Fig. 6c. Indeed, in this figure, you have chosen the green color also for subsidence rates from 0 to 10 mm/years.**

**Response:** The subsidence rate obtained by InSAR is characterized by an uncertainty of 1-3 mm/y (Teatini et al., 2012). This has been added in the ms. Considering the subsidence data used in this study has an error of ±7mm (for one year), the threshold of subsidence rate was set to 10 mm/y. Figure 6c showed the change of subsidence rate not the subsidence rate, in which the green color represents the decrease of subsidence rate.

**(9) Table 1: It could be interesting if you would provide in the discussion section some considerations about the factors which resulted in the highest difference between the computed fuzzy and no-fuzzy weights.**

**Response:** Added. We found that the stronger semantic fuzziness of the factor importance, the greater uncertainty of factor weights, which may make the greater difference in hazard probability results. This has been added in Section 4.2.

**(10) Line 212: Does it means that, differently for 2011-2014 and 2015-2017, for the period 2003-2010 the prior and posterior probability are computed on the same InSAR data? Could these make a difference in the three resulting hazard probability maps?**

**Response:** For period 2003-2010, the posterior probability was computed from the InSAR data. Due to the lack of subsidence information before 2003, the prior probability was computed based on the ratio of the feature grid number, which was introduced in line 207.

**(11) Line 226: Is that right that SrD (i.e. subsidence rate decrease) is located in the region with the highest groundwater level decrease between 2011-2014 and 2015-2017? I would say that where the subsidence rate decreased between these two periods, the groundwater level was restored accordingly, thus the area is SrD, and in a lower hazard class for 2015-2017. If I misinterpreted what you wanted to communicate, I suggest clarifying this part by using shorter statements.**

**Response:** It's right that points with SrD (i.e. subsidence rate decrease) is generally located in the region with the highest groundwater level recover between 2011-2014 and 2015-2017. However, Table 1 showed that some points with SrD are located in high hazard area. The reason is that there are portions of the study plain where, because the piezometric level did not recover significantly, land subsidence rates remained larger than 50 mm/y in 2017 (Zhu et al. 2020a, b) although smaller than the values observed in previous years. Although the subsidence rate decreased, these areas still have high hazard.

**(12) Line 245-247 and linked Fig. 8: Could you provide an explanation (also with the help of previous works) for why after a peak of hazard probability is reached both for compressible layer and Quaternary layer at a certain thickness values and then, for higher thicknesses, the hazard probability is much more lower? In my opinion, an interesting discussion on this could be done.**

**Response:** The thick compressible layers are prone for the occurrence of land subsidence. However, the drawdown of piezometric head is the triggering factor of land subsidence. The maximum subsidence occurred in the area where groundwater level dropped more rapidly, corresponding to a medium value of compressible layer and Quaternary layer thickness (160m-180m and 400m-600m, respectively).

**(13) Line 288: As I have previously mentioned in the comments for the Introduction section, I would suggest clarifying the statement regarding the early warning system.**

**Response:** Suggestion followed.

**(14) Technical Comments Line 37 and Line 41: The use of "the" before land subsidence should be avoided. Line 41: The use of "but" in the statement is unnecessary. Line 55: I would recommend changing "validity of the data" into "data quality". Line 89: I would suggest changing "technology" into "processing".**

**Response:** Corrected. Thanks for the suggestion.

**(15) Line 104-105: The meaning of Y is not specified. Moreover, I think there is a mistake in the statement: "P(Y|S) is the posterior probability of Y subject to S". S is land subsidence, thus it is the phenomenon for which you want to calculate the probability influenced by the factor Y.**

**Response:** Corrected. Y represents the hazard factor, which has been added in the ms. The definition of $P(Y|S)$ is revised to "the posterior probability of $Y$ when $S$ was observed".

**(16) Line 129-130: You stated that the FWBM method contains three parts, but you just enumerated one, so the statement is incomplete. Figures: measurement units in figure 6c and 8 are missing.**

**Response:** The missing statement is completed. FWBM contains three parts including probability of $Y_j$ which is consists of its spatial feature $X_{j,i}$, fuzzy weight of $Y_j$ and probability of $T$. The units in figure 6c and 8 were added.

[Figure]

**Fig. 6. (a) Assessment of the subsidence hazard probabilities from 2003 to 2010, 2011 to 2014, and 2015 to 2017; (b) Subsidence hazard level (2015-2017); (c) Change in land subsidence rate between 2010-2014 and 2015-2017 obtained by InSAR**

[Figure]

[Figure]

**Fig. 8. Hazard probability of factors for the three time periods from 2003 to 2017 of the study area**

**(17) About the English language, I think it needs some improvements (singular/plural and punctuation need to be checked, tenses need to be homogenised). If possible, I suggest a professional proofreading.**

**Response:** The English language is carefully checked and the errors were corrected. The manuscript was sent for professional proofreading at AJE, and carefully edited by the co-author Prof. Pietro Teatini.

---

## Author Comment (AC2) · 5 Jan 2021

<h1 style="text-align:center">Responses to the comments of Referee 2</h1>

We appreciate the positive and constructive comments provided by the Referee 2. We have revised the manuscript carefully by incorporating those comments. The following are our responses to the referee comments.

**(1)  The author needs to revise the English grammar, which is confused in some sections.**

**Response:** The English grammar is carefully checked and the errors were corrected. The singular/plural, punctuation tenses were checked and corrected. Some descriptions in the ms were revised for a better understanding. The ms was sent for professional proofreading at AJE, and carefully edited by co-author Prof. Pietro Teatini.

[Figure]

**(2)  In this work, some operations need to be more explaining for a better understanding of this approach.**

**Response:** We checked the method description again.

As for Section 2 (Methodology):

In line 105, the definition of $Y$, $Y$ represents the hazard factor, was added in the ms. The definition of $P(Y|S)$ is revised to "the posterior probability of $Y$ when $S$ is observed".

In line 110, the definition of $Y_j$ is revised as "$Y_j$ is the $j$-th of the $m$ factors that influence $T$".

In line 121, the description of Equation (3) is revised as "We developed the FWBM by extend Equation (2) with the introduction of a fuzzy-based weight".

In line 128, the definition of $X_{j,i}$ is revised as "$X=\{X_{j,1}, X_{j,2}, ..., X_{j,i-1}, X_{j,i}, ..., X_{j,n}\}$, where $X_{j,i}$ is defined as the $i$-th of the $n$-th features of the $j$-th factor. The value of $n$ depends on the feature classification.".

In line 134, the definition of prior probability "with the initial value calculated with feature grid number ratio" is unclear. It is revised as "The initial prior probability is calculated based on the feature grid number ratio between the number of grid cell with that feature and the total number of grid cells covering the study area.".

Section 3 (Case study):

In line 186, the description "the rate of decline in the groundwater level slowed due to the long-term loss of groundwater and increased rainfall (Zhang et al. 2015)" is not clear. This is revised as "the rate of decline in the groundwater level slowed due to the long-term loss of groundwater which reduces the capacity of water supplying and increased rainfall (Zhang et al. 2015)".

In line 192, the PS points with a subsidence rate above 10 mm/y were regarded as subsidence points. The reason for the threshold is explained as follows. The subsidence rate obtained by InSAR is characterized by an uncertainty of 1-3 mm/y, depending on the number and quality of the processed images (Teatini et al., 2012). Considering the subsidence data used in this study has an error of ±7mm (for one year), the threshold of subsidence rate was set to 10 mm/y.

Section 4 (Results and discussion)

In line 231, the levels of change between FWBM and WBM were calculated (the numbers in the legend in Fig. 7), which is not clear. This has been explained in Section 3.3.1. The result of WBM was reclassified and subtracted it from FWBM to compare the levels of change while consider or not consider the ambiguity.

In Table 2, the calculation of the match ratio is explained in the caption. The match ratio is calculated by the ratio between the sum of the amount of SrI in high hazard area and SrD in low hazard area, and the total number of SrI and SrD.